Morphological variation of leaf traits in the Ternstroemia lineata species complex (Ericales: Penthaphylacaceae) in response to geographic and climatic variation

Alcántara-Ayala Othón 1 2
Oyama Ken 3
Ríos-Muñoz César A. 4
Rivas Gerardo 5
Ramirez-Barahona Santiago 6
Luna-Vega Isolda isolda_luna-vega@ciencias.unam.mx 2
1 Posgrado en Ciencias Biológicas, Universidad Nacional Autónoma de México , Mexico City , México
2 Laboratorio de Biogeografía y Sistemática, Departamento de Biología Evolutiva, Facultad de Ciencias, Universidad Nacional Autónoma de México , Mexico City , México
3 Escuela Nacional de Estudios Superiores (ENES), Unidad Morelia, Universidad Nacional Autónoma de México , Morelia , Michoacán , México
4 Coordinación Universitaria para la Sustentabilidad, Universidad Nacional Autónoma de México , Mexico City , México
5 Departamento de Biología Comparada, Facultad de Ciencias, Universidad Nacional Autónoma de México , Mexico City , México
6 Departamento de Botánica, Instituto de Biología, Universidad Nacional Autónoma de México , Mexico City , México
Schietti Juliana
Electronic publication date: 2020 Jan 10
Publication date: 2020
Volume: 8
Electronic Location ID: e8307
Received 2019 Mar 30; Accepted 2019 Nov 27
Copyright: ©2020 Alcántara-Ayala et al.
Copyright year: 2020
Copyright holder: Alcántara-Ayala et al.
License: This is an open access article distributed under the terms of the Creative Commons Attribution License, which permits unrestricted use, distribution, reproduction and adaptation in any medium and for any purpose provided that it is properly attributed. For attribution, the original author(s), title, publication source (PeerJ) and either DOI or URL of the article must be cited.
License URL: https://creativecommons.org/licenses/by/4.0/

Keywords: Morphological variation, Ternstroemia, Leaf traits, Geographical variation, Species complex

Funding: Consejo Nacional de Ciencia y Tecnología 245683 Dirección General de Asuntos del Personal Académico-Programa de Apoyo a Proyectos de Investigación e Innovación Tecnológica IV201015 IN215914 This work was supported by the Consejo Nacional de Ciencia y Tecnología (PhD grant for Othón Alcántara-Ayala No. 245683) and the Dirección General de Asuntos del Personal Académico-Programa de Apoyo a Proyectos de Investigación e Innovación Tecnológica (No. IV201015 and No. IN215914). The funders had no role in study design, data collection and analysis, decision to publish, or preparation of the manuscript.

==============================
Variation in leaf morphology is correlated with environmental variables, such as precipitation, temperature and soil composition. Several studies have pointed out that individual plasticity can largely explain the foliar phenotypic differences observed in populations due to climatic change and have suggested that the environment plays an important role in the evolution of plant species by selecting for phenotypic variation. Thus, the study of foliar morphology in plant populations can help us identify the environmental factors that have potentially influenced the process of species diversification. In this study, we analyzed morphological variation in the leaf traits of the Ternstroemia lineata species complex (Penthaphylacaceae) and its relation to climatic variables across the species distribution area to identify the patterns of morphological differentiation within this species complex. Based on the collected leaves of 270 individuals from 32 populations, we analyzed nine foliar traits using spatial interpolation models and multivariate statistics. A principal component analysis identified three main morphological traits (leaf length and two leaf shape variables) that were used to generate interpolated surface maps to detect discrete areas delimited by zones of rapid change in the values of the morphological traits. We identified a mosaic coarse-grain pattern of geographical distribution in the variation of foliar traits. According to the interpolation maps, we could define nine morphological groups and their geographic distributions. Longer leaves, spatulate leaves and the largest foliar area were located in sites with lower precipitation and higher seasonality of precipitation following a northwest–southeast direction and following significant latitudinal and longitudinal gradients. According to the phenogram of the relationships of the nine morphological groups based on morphological similarity, the putative species and subspecies of the T. lineata species complex did not show a clear pattern of differentiation. In this study, we found a complex pattern of differentiation with some isolated populations and some other contiguous populations differentiated by different traits. Further genetic and systematic studies are needed to clarify the evolutionary relationships in this species complex.

Introduction

Intraspecific variation is one of the main sources of information for recognizing evolutionary patterns. Identifying the causes of intraspecific variation is essential to understanding the evolutionary processes that maintain diversity and promote speciation (Futuyma, 1998). In plants, variation occurs in populations of species that are separated in space, encompassing both the genotype and phenotype (Thorpe, 2002). The causes of phenotypic variation among individuals across the geographical distribution range of a species can be broadly divided into current environmental conditions within particular habitats and historical processes and phylogenesis (Thorpe, 1987; Peppe et al., 2011). Plant populations of the same species growing under different environmental conditions respond to different selection pressures, producing genetic and phenotypic divergence between populations (Ramsey, Cairns & Vaughton, 1994; Fenster & Stenøien, 2001; Albarrán-Lara et al., 2019).

Plants have developed particular adaptations to the surrounding local climate that allow them to be better fitted to their environment (Chevin, Lande & Mace, 2010; Valladares et al., 2014). The leaf is the structure in which changes in morphology in response to their environment are more readily acquired by plants (Malhado et al., 2009a; Yang et al., 2015) because leaves are the organs that perform essential functions, such as photosynthesis and regulation of water content (Givnish, 1979; Wright et al., 2004; Adams & Ichiro, 2018; Tsukaya, 2018).

Latitudinal and altitudinal variations in leaf size and shape are the result of plastic and adaptive responses of plants to varying environmental conditions, with differences in plant physiological responses also present among populations (Rico-Gray & Palacios-Ríos, 1996; Niinemets, 2001; Uribe-Salas et al., 2008; Frenne et al., 2013; Moles et al., 2014). The correlation between the shape and size of leaves and the surrounding habitat has been interpreted as the result of an evolutionary response of plants to varying environmental conditions (Givnish, 1987; Westoby et al., 2002; Peppe et al., 2011). For instance, leaf size has been shown to decrease with increasing altitude because of the differences in precipitation and with the decrease of soil nutrient content (McDonald et al., 2003; Nicotra et al., 2011). Additionally, smaller leaves appear to be better adapted to dry environments because the smaller size reduces the hydraulic vulnerability of leaves and makes the plants more tolerant to drought (Scoffoni et al., 2011). Light intensity also plays a key role in the adjustment of leaf size and shape in later stages of development, promoting the expansion of the leaf petiole and inhibiting the growth of the leaf blade (Tsukaya, 2005).

Several environmental factors have been shown to play key roles in the evolution of plant species by selecting for phenotypic variation (Pfennig et al., 2010), especially variation related to leaf traits. Thus, the study of foliar morphology in plant populations can help us to identify the environmental factors that have potentially influenced the process of species diversification and phenotypic variability (Aschcroft, French & Chisholm, 2011). The relationships of leaf variation with environmental variables have been focused on the global (Ordoñez et al., 2009; Yang et al., 2015; Wright et al., 2017), regional (e.g., North America) (Royer et al., 2008) and local (e.g., South Australia, Amazonia and Bolivia) (Sokal, Crovello & Unnasch, 1986; Wright & Ladiges, 1997; Gregory-Wodzicki, 2000; Malhado et al., 2009a; Malhado et al., 2009b; Guerin, Wen & Lowe, 2012) scales. However, similar patterns are not always identifiable across spatial scales, and in some cases, contrasting trends can be found (Malhado et al., 2009b; Wright et al., 2017). Studies on how plants respond to environmental variables within regions with a high degree of spatial heterogeneity could be useful for understanding the differential response of plants to climatic gradients (Moeller & Merilä, 2004). In turn, this knowledge would allow us to know the degree and direction of evolutionary divergence between populations.

The genus Ternstroemia (Penthaphylacaceae, order Ericales) is a group of plants that is poorly understood taxonomically, with the total number of species accepted in the genus varying from 90–110 species (Stevens, 2001; Weitzman, Dressler & Stevens, 2004; Xiang, 2007) to over 160 species (Weitzman, 1995). The genus is distributed from Sri Lanka to SE and E Asia and exists in tropical and subtropical America and in Africa (two species) (Weitzman, Dressler & Stevens, 2004). In Mexico and Central America, Ternstroemia can be found in regions with a great variety of climates and habitats, ranging from 50 m to more than 3,000 m (Luna-Vega, Alcántara-Ayala & Contreras-Medina, 2004). One of the most widely distributed and common species of Ternstroemia in Mexico is Ternstroemia lineata. In particular, the taxonomic circumscription of T. lineata has been difficult and thus is considered a species complex composed of several taxa: T. lineata subsp. chalicophila (Loes.) B.M. Barthol., T. dentisepala B.M. Barthol., T. lineata subsp. lineata and T. impressa Lundell. The T. lineata species complex is distributed exclusively in Mesoamerica and is restricted to cloudy conditions at elevations ranging from 1,400 to 3,140 m in habitats characterized by high precipitation (>1,000 mm annually), such as cloud forests and mixed humid forests (Alcántara-Ayala, Luna-Vega & Velázquez, 2002; Luna-Vega, Alcántara-Ayala & Contreras-Medina, 2004). In Mexico and Guatemala, these types of forests occur within a relatively narrow altitudinal zone under humid, temperate climates with continuously foggy conditions. However, different forest patches show local climatic variations that generate heterogeneity in floristic composition. Accordingly, individuals in the T. lineata species complex show great variation in leaf shape and size among populations (Kobuski, 1942; Bartholomew & McVaugh, 1997; González-Villarreal, 2001).

In this study, we used the Ternstroemia lineata complex as a model to test how climates influence foliar morphology in forest species under environments with high humidity but great geographic heterogeneity. We analyzed how leaf morphology of the T. lineata species complex vary in relation to climate variables across its geographic distribution to detect which environmental variables have influenced the leaf morphological differentiation in this species complex. We also tested the degree of leaf morphological differentiation among the putative species and subspecies complex of T. lineata.

Materials & Methods

Taxon sampling and morphological trait analysis

We collected leaves of 270 individuals (8–10 mature leaves per individual) from 32 populations located throughout the distribution range of the species (Fig. 1). A field permit of scientific collection (Official number SGPA/DGVS/12770/16) was issued by the Secretaria de Medio Ambiente y Recursos Naturales, of Mexico. Leaf samples were pressed and dried for further measurements in the lab and for herbarium specimens. For each specimen, we measured nine foliar traits: 1. total length including lamina and petiole (TL); 2. lamina length (LL); 3. maximum width of the lamina (MW); 4. petiole length (PL); 5. distance from the base to the maximum width of the lamina (DW); 6. petiole diameter (PD); 7. angle of the lamina apex (ALA); 8. ratio between MW and LL (WLR); and 9. ratio between DW and LL (DWLR) (Fig. 2A). All variables (except ALA) were measured using a Mitutoyo® Vernier caliper (0.05 mm resolution) and are recorded in mm. For ALA, we used a Jeppesen PJ-1 Rotating Azimuth Plotter.

Variation of morphological traits

We performed a principal component analysis (PCA) with the nine traits using the software Statistica (Statsoft Inc, 2009) to group the variables according to their variability and to select those with the higher values per component to account for most of the morphological variation among individuals (Table 1). With the selected morphological traits (TL, WLR and DWLR), interpolated surface maps were generated in a geographic information system (GIS) (ESRI, 2011) using the Geostatistical Analyst extension (Johnston et al., 2001). The traits DWLR and WLR are descriptors of leaf shape. DWLR refers to the degree to which a leaf is spatulate or elliptical, and WLR is indirect evidence of the foliar area (Nautival et al., 1990; Cittadini & Peri, 2006; Singh, 2007), where high values indicate small foliar areas and low values indicate large areas.

Figure 1 Location of the sampled populations of the T. lineata complex.

Distribution of the Ternstroemia lineata species complex in the main mountain systems of Mexico (sensu INEGI, 2001) and northern Central America (sensu Marshall, 2007).

Figure 2 Nine foliar traits and Principal Component Analysis (PCA) of this variables.

Foliar measurements and PCA analysis of the morphological variables. (A) The nine foliar traits measured in each individual: (1) total leaf length (including lamina and petiole, TL); (2) lamina length (LL); (3) maximum width of the lamina (MW); (4) petiole length (PL); (5) distance from the base to the maximum width of the lamina (DW); (6) petiole diameter (PD); (7) angle of the lamina apex (ALA); (8) ratio between MW and LL (WLR); (9) ratio between DW and LL (DWLR). (B–C) We plotted the PCA values of the morphological variables: first component vs. second component (B) and second component vs. third component (C) we show two sets of variables with maximum correlation: LL-TL-DW and ALA-WLR. The number (B and C) corresponds to species of T. lineata complex, 1. T. lineta ssp. chalicopila, 2. T. dentisepala, 3. T. lineata ssp. lineata and 4. T. impressa.

Table 1 Climatic variables used in the redundancy analysis.

Abbreviations	Climatic variable names	
BIO1a	Annual Mean Temperature	
BIO2	Mean Diurnal Range (Mean of monthly (max temp - min temp)	
BIO3	Isothermality (BIO2/BIO7) (* 100)	
BIO4a	Temperature Seasonality (standard deviation *100)	
BIO5	Max Temperature of the Warmest Month	
BIO6	Min Temperature of the Coldest Month	
BIO7	Temperature Annual Range (BIO5-BIO6)	
BIO8	Mean Temperature of the Wettest Quarter	
BIO9	Mean Temperature of the Driest Quarter	
BIO10	Mean Temperature of the Warmest Quarter	
BIO11	Mean Temperature of the Coldest Quarter	
BIO12	Annual Precipitation	
BIO13a	Precipitation of the Wettest Month	
BIO14a	Precipitation of the Driest Month	
BIO15a	Precipitation Seasonality (Coefficient of Variation)	
BIO16	Precipitation of the Wettest Quarter	
BIO17	Precipitation of the Driest Quarter	
BIO18	Precipitation of the Warmest Quarter	
BIO19a	Precipitation of the Coldest Quarter	
Notes.

a Climatic variables with VIF > 10.

Finally, the interpolated maps were clipped with a map of the selected hydrographic basins of Mexico (INEGI, INE & CONAGUA, 2007) based on the presence of T. lineata complex records and the altitudinal range obtained from the herbarium specimens and field collections to eliminate the interpolation areas outside the known distribution of the T. lineata complex.

We used an empirical Bayesian kriging method, which is based on a semivariogram estimated from the spatial structure of the data (Oliver & Webster, 1990; Kidd & Ritchie, 2006; Brito et al., 2008). This kriging method is a geostatistical tool that generates an estimated surface from a set of dispersed points with Z values. The method is based on statistical models that consider the spatial autocorrelation among data points and provide a measure of precision of the predicted values. We considered the mean error, the square root of the quadratic mean error, the mean standard error, and the square root of the quadratic mean error to evaluate the efficiency of the interpolation (Johnston et al., 2001).

Geographical areas of morphological homogeneity

The three interpolated maps were used to detect discrete areas delimited by zones of rapid change in the values of morphological traits because the variation inside the area is lower than that among the areas. For this method, we used a boundary delineation method with the BoundarySeer software (settings used: crisp boundary type, top 20% in boundary in threshold(s), thresholds 90 deg. vector-to-vector and 30 deg. vector-to-line gradient angle). Different boundaries are zones of rapid change, and for detection, wombling methods are used, which estimate the average amount of change in the variable across the space. The locations that have high values of change are referred to as boundaries (BioMedware, 2013).

Each map generated by the boundary analysis was vectorized, and all were intersected in a single map using ArcGIS Ver. 10.0 (ESRI, 2011). The resulting polygons represented areas of geographical coincidence of the distribution of the three morphological traits: those polygons where the presence of T. lineata complex was not corroborated were eliminated based on the records of biodiversity information systems (REMIB, GBIF, specimens of herbaria). To reduce the number of polygons and group them into homogeneous areas, a cluster analysis was carried out based on the mean values of each of the morphological traits present in each polygon. This analysis was performed using NTSYSpc ver 2.11 (Rohlf, 2000) with a taxonomic distance coefficient and the UPGMA algorithm, and group formation was taken at a value of 0.88. Finally, the resulting groups were evaluated using discriminant analysis to identify their differences (Table 2). Finally, the distributions of the values of each of the three variables within each group were represented using boxplots. Discriminant analysis and boxplot were carried out using SPSS Statistics ver. 19 software (IBM Corp. Released, 2010).

Table 2 Principal Component Analysis of nine morphological variables of the T. lineata species complex.

Variables: (1) total length of leaf (includes lamina and petiole, TL); (2) lamina length (LL); (3) maximum width of the lamina (MW); (4) petiole length (PL); (5) distance from the base to the maximum width of the lamina (DW); (6) petiole diameter (PD); (7) angle of the lamina apex (ALA); (8) ratio between MW and LL (WLR); and (9) ratio between DW and LL (DWLR).

Contributions	C1	C2	C3	C4	C5	
PL	0.041737	0.000777	0.161636	0.726383	0.016466	
PD	0.054816	0.186583	0.009767	0.068551	0.258501	
MW	0.189874	0.083223	0.015169	0.012927	0.000201	
TL	0.244426	0.029179	0.002038	0.001309	0.011538	
LL	0.239342	0.032236	0.000070	0.018755	0.016813	
DW	0.228145	0.028064	0.052958	0.000150	0.001161	
WLR	0.000166	0.379410	0.026536	0.000723	0.038457	
DWLR	0.000956	0.000000	0.663112	0.148503	0.081381	
ALA	0.000538	0.260529	0.068714	0.022698	0.575481	
% ACUMULATED VARIANCE	41.83	66.15	79.71	94.04	99.59	

Relationship of morphological traits with climate and geography

To evaluate the relationship of the morphological traits with the climate across the range of the distribution of this species complex, we used the average values of each polygon, considering 19 bioclimatic variables (Hijmans et al., 2005) (Table 1) and three geographical variables. Values were obtained through a zonal analysis in the GIS. We used the redundancy analysis (RDA) implemented in the package ‘vegan’ (Oksanen et al., 2018) in R 3.5.1 software (R Core Team, 2018) to determine the combination of environmental and geographical variables that better explain the morphological variation. This analysis was carried out as follows: first, all the bioclimatic variables (BIO01 to BIO19) were included, and an iterative analysis was carried out so that the variables showing high collinearity were eliminated according to their values of variance inflation factor (VIF). Then, to evaluate the effects of climate and geography on leaf trait variation, we performed RDA analysis in three phases: (a) an analysis that includes both climatic and geographical variables (RDAfull); (b) a partial RDA analysis including only bioclimatic variables without collinearity (pRDA1); and finally, (c) a partial RDA analysis including only geographical variables (pRDA2).

We also conducted linear Pearson correlation analyses between the three morphological traits selected after the PCA and the bioclimatic variables without collinearity after an iterative analysis. We included geographic variables such as latitude, longitude and altitude in these linear correlation analyses to detect patterns of leaf morphological variation through latitudinal, longitudinal or altitudinal gradients.

Results

Leaf morphological variation and interpolation maps

The principal component analysis of the morphological traits showed that the first three components explained 79.71% of the accumulated variance (41.83%, 24.32% and 13.56%). For the first component, the higher loading values were associated with leaf length (TL and LL) and width (DW and MW), and the second and third components were associated with leaf shape (WLR and DWLR, respectively) (Table 2). The relation among the components showed two sets of variables with maximum correlation associated with the leaf length and width (LL, TL and DW) and the lamina angle and leaf shape (ALA and WLR) (Figs. 2B and 2C).

For the three variables identified as important for each component (TL, WLR and DWLR), we obtained the interpolated surface maps (Figs. 3A–3C). We identified a coarse-grain geographical mosaic distribution in the variation of these foliar traits. The longest leaves occurred in southern Mexico (Sierra Madre del Sur), and the plants with the shortest leaves were mainly located in the southernmost region of the distribution (Sierra Norte y Los Altos de Chiapas and Sierra Madre de Chiapas and Cuchumatanes) and a small area in the western part of the Eje Neovolcánico (Fig. 3A). Plants with spatulate leaves (higher values of DWLR) were mainly located in the western part of the Eje Neovolcánico and in the southernmost part of the Sierra Madre de Chiapas and Cuchumatanes. Plants with elliptical leaves (lower values of DWLR) were located mainly in the Sierra Madre del Sur and Sierra del Norte y Altos de Chiapas (Fig. 3B). The regions with plants with the largest foliar area (low values of WLR) were mainly located in the Sierra Madre del Sur, Eje Neovolcánico and Sierra Madre Occidental. The highest WLR values (smaller foliar area) are located in Sierra Madre de Chiapas and Cuchumatanes (Fig. 3C).

Geographical areas of morphological homogeneity and the morphological delimitation of the T. lineata species complex

The boundary analyses and the vectorization of the boundary maps produced 228 polygons for the TL map, 395 polygons for the DWLR, and 270 for the WLR (Figs. 3D–3F). The intersection of these three maps and the subsequent elimination of polygons, using only the records of presence of the T. lineata complex, produced a single map with 108 polygons (Fig. 4A). The cluster analysis with the average values of the three morphological traits in these 108 polygons generated nine clusters (morphological groups) (Fig. 4B). The discriminant analysis showed significant differences among these nine clusters (Table 3) with the eigenvalues of the first two functions discriminated by 92.7% with small Wilks’ lambda values, which confirms the high variability among these groups. Finally, the percentage error among the groups was minimal (0.9%) (Fig. 5A). Figure 5B shows the differences of the three morphological traits among the nine morphological groups.

Figure 3 The interpolated surface maps.

Interpolation maps for (A) total leaf length (TL), (B) ratio between the distance from the base to the maximum width of the lamina (DW) and leaf length (LL) (DWLR) and (C) ratio between maximum width of the lamina (MW) and leaf length (LL) (WLR). (D–E) Boundary detection analyses based on the interpolated surface maps of the three leaf traits (TL, DWLR, WLR, respectively).

Figure 4 The intersection of three maps and phenogram resulted from a cluster analysis of the 108 polygons.

(A) Map with polygons generated by the intersection of the three maps of morphological variables (see Figs. 3D–3F) after the boundary analysis and the vectorization of the boundary maps using only the records of presence of the T. lineata complex (green dots). (B) Phenogram resulted from a cluster analysis of the 108 polygons.

Table 3 The discriminant analysis showing significant differences among the nine clusters.

(A) Eigenvalues of the functions show that the first two functions discriminate by 92.7%. (B) Wilks Lambda values and Chi-square, p = 0.05.

Function	Eigenvalue	% of variance	Cumulative %	Canonical correlation	
(A) Eigenvalue	
1	10.251a	60.3	60.3	.955	
2	5.506a	32.4	92.7	.920	
3	1.249a	7.3	100.0	.745	
Test of functions	Wilks’ Lambda	Chi-square	df	Sig	
(B) Wilks’ Lambda	
1 to 3	0.006	515.478	24	.000	
2 to 3	0.068	271.010	14	.000	
3	0.445	81.858	6	.000	
Notes.

a We used the three first canonical discriminant functions in the analysis.

Figure 5 Biplot discriminant analysis, boxplot of the differences of the three morphological traits among the nine morphological groups and distribution map of this nine morphological groups.

(A) Canonical discriminant charts where the integrity of the nine groups was tested; (B) boxplot of the nine morphological groups; (C) distribution map of the nine morphological groups (with a typical leaf sample).

The nine morphological groups have some correspondence with the species and subspecies of the T. lineata species complex. T. lineata subsp. lineata were formed by the populations of group 1 but also by populations of groups 3, 7 and 9 and some populations of group 4 (in allopatry) and other populations of groups 4 and 5 (in parapatry). Populations of group 2 belong to T. dentisepala, which is distributed in Nayarit state (northwest Mexico), populations of group 6 belong to T. lineata subsp. chalicophila (Chiapas state in southern Mexico) and populations of group 8 belong to T. impressa, which is distributed in Guatemala (Fig. 5C). However, according to the phenogram of the relationships of the nine morphological groups based on morphological similarity, the putative species and subspecies did not show clear patterns of differentiation. For example, the morphological similarity within morphological group 2 (T. dentisepala) was higher and was also higher within the morphological groups of T. lineata subsp. lineata (groups 1, 3 and 4). Group 5 was an interesting case because the populations of this group were distributed parapatrically and differentiated from the main cluster of T. lineata subsp. lineata. Populations of the morphological groups 6, 7, 8 and 9 were geographically isolated and differentiated in different clusters, and plants of group 8 that were distributed in Guatemala were described as T. impressa (Fig. 5C).

Relationship of geography and climate to morphological traits

We selected six bioclimatic variables (BIO01, BIO04, BIO13, BIO14, BIO15 and BIO19) based on an iterative analysis. The results of the full redundancy analysis (F = 10.037, p < 0.001, N = 999 permutations), including geographic and bioclimatic variables, explained 47.79% of leaf variation (p < 0.001) with two significant axes RD1 and RD2; the first axis explained 83.72% of the variance, and the second axis explained 13.61%. The most significant variables for the first axis were precipitation of the driest month (BIO14, p < 0.001), precipitation seasonality (BIO15, p < 0.043), precipitation of the coldest quarter (BIO19, p < 0.007) and longitude (p < 0.001); for the second axis, the most significant variables were latitude (p < 0.001), precipitation of the wettest month (BIO13, p < 0.024), temperature seasonality (BIO04, p < 0.014) and annual mean temperature (BIO01, p < 0.001). The most important morphological variables were total leaf length (TL) and width-length ratio (WLR). The first axis of the full RDA separated those populations mainly located in the south in Sierra del Norte y Los Altos de Chiapas and Sierra Madre de Chiapas and the populations of the Sierra Madre del Sur (in the state of Oaxaca) and the most western area of the Eje Neovólcanico (state of Nayarit) (Fig. 6).

Figure 6 Graph of the redundancy analysis that includes both climatic and geographical variables (RDAfull).

Graph of full redundancy analysis (Full RDA) showing the relationship between response variables (morphological characters) and explanatory variables (bioclimatic and geographic variables).

The analysis of leaf trait variation indicated that climate had a higher impact than geography. The percentage of the variance explained only by climate was 31.08%, the percentage explained by geography was 13.38% and the join effect, geography—climate, explained 2.38%.

The partial redundancy analysis using climate as control of geography (pRDA1) was significant only in its first axis (p < 0.001), explaining 87.26% of the variation, with precipitation of the coldest quarter (BIO19, p < 0.009) as the most important variable. The partial redundancy analysis using geography as a control climate (pRDA2) was only significant in its first axis (p < 0.001), which explains 99.45% of the variation, with latitude being the variable of greater importance (p < 0.001) (Fig. 6).

Relationship between morphological traits and environmental and geographic variables

After the principal component analysis defined TL, WRL and DWLR as the morphological traits that explain most of the variance in the T. lineata species complex and the iterative analysis selected the climatic variables without collinearity, we conducted linear correlations between both groups of variables. Total leaf length (TL) was positively correlated with annual mean temperature (BIO01) (r = 0.30; p < 0.01) and precipitation seasonality (BIO15) (r = 0.48; p < 0.01) and negatively correlated with the precipitation of the driest month (BIO14) (r =  − 0.53; p < 0.01), precipitation of the wettest month (BIO13) (r =  − 0.13; p < 0.01) and precipitation of the coldest quarter (BIO19) (r = 0.45; p < 0.01) (Fig. 7).

Figure 7 Correlations between total leaf length and bioclimatic variables.

(A) Correlations between total leaf length (TL) and bioclimatic variables were positively correlated with annual mean temperature (BIO01) and (B) precipitation seasonality (BIO15) and (C) negatively correlated with precipitation of the driest month (BIO14), (D) precipitation of the wettest month (BIO13) and (E) precipitation of the coldest quarter (BIO19).

The ratio between the maximum width of the lamina (MW) and leaf length (LL) (WRL) as a descriptor of leaf shape was positively correlated with the precipitation of the wettest month (BIO13) (r = 0.39; p < 0.01) and precipitation of the driest month (BIO14) (r = 0.31; p < 0.01) and was negatively correlated with temperature seasonality (BIO04) (r =  − 0.27; p < 0.01) and precipitation seasonality (BIO15) (r =  − 0.37; p < 0.01) (Fig. 8).

Figure 8 Correlations of the ratio between maximum width of the lamina/leaf length (WRL) and bioclimatic variables.

The ratio between maximum width of the lamina (MW) and leaf length (LL) (WRL) as descriptor of leaf shape was (A) positively correlated with precipitation of the wettest month (BIO13) and (B) precipitation of the driest month (BIO14) and (C) negatively correlated with temperature seasonality (BIO04) and (D) precipitation seasonality (BIO15).

The ratio between the distance from the base to the maximum width of the lamina (DW) and leaf length (LL) (DWLR) was positively correlated with temperature seasonality (BIO04) (r = 0.40; p < 0.01) and precipitation seasonality (BIO15) (r = 0.25; p < 0.05) and negatively correlated with the precipitation of the wettest month (BIO13) (r =  − 0.21; p < 0.05), precipitation of the driest month (BIO14) (r = 0.32; p < 0.01) and precipitation of the driest quarter (r =  − 0.41; p < 0.01) (Fig. 9).

Figure 9 Correlations between the ratio between the distance from the base to the maximum width of the lamina/leaf length (DWLR) and bioclimatic variables.

The ratio between the distance from the base to the maximum width of the lamina (DW) and leaf length (LL) (DWLR) was (A) positively correlated with temperature seasonality (BIO04) and (B) precipitation seasonality (BIO15) and (C) negatively correlated with precipitation of the wettest month (BIO13), (D) precipitation of the driest month (BIO14) and (E) precipitation of the driest quarter.

Additionally, TL and DWLR were positively correlated with latitude (r = 0.20; p < 0.05; r = 0.22; p < 0.05, respectively) and negatively correlated with longitude (r =  − 0.30; p < 0.05; r =  − 0.24; p < 0.05, respectively), but WLR was negatively correlated with latitude (r =  − 0.53; P < 0.01) and positively correlated with longitude (r = 0.49; p < 0.01). The three morphological traits were not correlated with altitude (TL: r =  − 0.12; p > 0.05; WLR: r =  − 0.08; p > 0.05; DWLR: r =  − 0.14; p > 0.05) (Fig. 10).

Figure 10 Correlations between morphological variables and latitude and/or longitude.

(A and B) TL and DWLR were positively correlated with latitude and (C and D) negatively correlated with longitude, (E) WLR was negatively correlated with latitude and (F) positively correlated with longitude.

Discussion

Here, we analyzed leaf morphology in the T. lineata species complex, a group of plants characteristic of the humid forests of Mesoamerica. To detect the role of environmental factors in morphological population differentiation, we studied the general geographic trends in leaf morphology and their relationships with climatic variables such as temperature and precipitation. The populations of the T. lineata species complex show great levels of variation in leaf morphology throughout their distribution range. Accordingly, our analyses allowed us to define nine morphologically homogenous groups within the complex and to define their geographic distributions. In general, the pattern we uncovered is a patchy mosaic geographic distribution with some groups of populations having a continuous distribution along the central and southern mountains of Mexico (Eje Neovolcánico, Sierra Madre del Sur). However, we also detected some morphological groups that are distributed parapatrically and others that are isolated at the northern and southern edges of the distribution range of the species complex. Although some groups are consistent with the present-day accepted taxonomic status (Kobuski, 1942; Bartholomew & McVaugh, 1997), such as the populations identified as T. dentisepala in the north and T. impressa in the south, other groups do not have a clear association with the currently accepted taxonomic circumscription.

Since the 19th century, biogeographers have found that wet tropics harbor plants with larger leaves than those observed in temperate regions, suggesting that small leaves are more frequently found at high latitudes and elevations. Recently, some authors (e.g., Jones, 2014; Michaletz et al., 2016; Wright et al., 2017) have attempted to explain the global climatic drivers of leaf size and shape. In general, morphological variation in plants follows changes in environmental variables along latitudinal, longitudinal and altitudinal gradients at different geographic scales (Rico-Gray & Palacios-Ríos, 1996; Niinemets, 2001; Chalcoff, Ezcurra & Aizen, 2008; Uribe-Salas et al., 2008; Frenne et al., 2013; Moles et al., 2014). In particular, complex leaf morphology in the T. lineata species showed clear patterns of latitudinal and longitudinal variation, but no patterns in morphology were detected across altitude. However, we uncovered strong correlations between leaf morphology and environmental variables. More specifically, leaf area showed a close association with environmental variables, where leaves with lower surface area were found in regions with higher precipitation throughout the year (i.e., high precipitation during the wettest and driest months), whereas leaves with higher surface area were found in regions with pronounced temperature and precipitation seasonality. Interestingly, this pattern does not follow the general trend observed in plants at a global scale, in which smaller leaves tend to occur in drier sites (Wright et al., 2017).

Variation in leaf shape along environmental gradients has been tied to the response of plant populations to environmental factors (Givnish, 1987; Malhado et al., 2009a; Malhado et al., 2009b; Werger & Ellenbroek, 1978). Accordingly, longer leaves in T. lineata were found in regions with higher temperatures and with strong precipitation seasonality, whereas the shortest leaves were found in regions with higher levels of rainfall. In general, longer, narrower and more spatulate leaves were located in regions with lower precipitation and higher precipitation seasonality, following a northwest—southeast direction. However, populations of the T. lineata species complex also show differentiation in leaf shape (i.e., narrow spatulate vs elliptical and obovate leaves) within similar mixed cloud forest habitats but still followed significant latitudinal and longitudinal gradients.

The relationship we observed between the length and the surface area of leaves with precipitation seasonality is difficult to explain. Traditionally, a reduction in leaf size has been reported for many species as aridity increases, which represents an adaptive advantage because small leaves have low evapotranspiration (Chalcoff, Ezcurra & Aizen, 2008). In contrast, our results show that individuals of the T. lineata complex with the longest leaves and with the largest leaf areas are found in regions with the lowest extreme rainfall. We believe that the pattern we are observing is due to the habitats in which these plants live. These habitats are characterized by high levels of humidity in the form of cloud, which is more persistent precisely in the driest and coldest periods of the year. In this case, the constant presence of cloud tends to form a thin layer of water on the surface of the leaf, which reduces transpiration and therefore growth (Leigh, 1975; Lightbody, 1985). In addition, this water film reflects sunlight, causing a reduction in photosynthetic activity and therefore affecting the growth rate of the leaves (Lightbody, 1985).

The adaptive value of having more spatulate or elliptical leaves with respect to varying environments is also difficult to explain. Recently, Kidner & Umbreen (2010) argued that leaf shape is extremely variable between and within species, with great levels of variation in leaf shape among populations and individual plants. The shape of the leaf has been related to the capacity to capture light within distinct habitats and to the regulation of water balance and temperature. In this case, a detailed study of microclimatic variation among populations and analyses of leaf vascular patterns are needed to determine the factors that drive the variation of leaf shape in the T. lineata complex.

Variation in leaf traits is due to phenotypic plasticity in response to environmental gradients, and the observed morphological variation is frequently strongly associated with climate (Givnish, 1987; Westoby et al., 2002; Gratani, 2014). In this context, the recognition of geographical and morphological groups along the distribution range of the T. lineata species complex helped us identify regions where environmental variation mirrored the observed morphological variation across populations. There is a clear mosaic-type geographical pattern of variation in some foliar traits, but there is also evidence of correspondence between environmental gradients and leaf size and shape. These results indicate that environmental factors play a relevant role in the observed variation in foliar traits in the T. lineata species complex. However, this trait variation is clustered within discrete groups of populations, which are well differentiated geographically. Interestingly, the observed patterns suggest that several groups of populations are morphologically differentiated as a result of geographic isolation. However, there is evidence of significant differentiation in leaf morphology among geographically contiguous populations, suggesting that factors other than climate might influence variation in leaf morphology in Ternstroemia.

Conclusions

We analyzed the relationship between environmental variables and variation in leaf morphology among populations throughout the distribution range of the T. lineata species complex in Mexico and Guatemala. We found that the effect of climate varies among populations and different morphological trends lead to varying patterns of geographic differentiation. The results indicate that converging leaf morphologies can be observed among individuals from different populations, which appears to be a parallel response to similar environmental factors rather than to geographical proximity. Although the climate-morphology association may eventually lead to adaptive evolution in this species complex, the observed patterns should be corroborated with analyses of genetic structure among populations.

The combination of multivariate statistical and geographic analyses of leaf morphology allowed us to establish variability patterns that are a fundamental first step to understanding the process of population differentiation in this group of closely related species. In turn, the integration of genetic data with morphological variation in vegetative and reproductive structures can lead to a better understanding of the reproductive barriers and the processes of species formation in these species. Our research highlights the key role of the environment in molding morphological variation among plant populations at the local and regional scales. Last, a formal taxonomic treatment that includes morphological traits of the leaf, flowers and fruits, together with molecular markers, is needed to evaluate the degree of differentiation among species and subspecies of the T. lineata complex.

Supplemental Information

Supplemental Information 1 Raw data used for PCA of nine foliar traits

Click here for additional data file.

Supplemental Information 2 Raw data used for generate interpolated surface maps

Click here for additional data file.

Supplemental Information 3 Raw data used for cluster and discriminant analysis

Click here for additional data file.

Supplemental Information 4 Raw data used for RDA anlaysis with climatic and geographic data

Click here for additional data file.

This paper is a requirement for O. Alcántara-Ayala in the graduate program of Posgrado en Ciencias Biológicas, Universidad Nacional Autónoma de México (UNAM). We acknowledge Andrés Torres-Miranda and Hernán Alvarado-Sizzo for field assistance and their critical suggestions on the final version of this manuscript. We thank to Celia Sanginés, Rito Vega, Antonio Solís, Francisco Maradiaga, Armando Ponce, Mario Véliz, Dagoberto Alavez and Andrés González for field assistance. We also thank to Ramón Cuevas for the facilities to access the Manantlán Biosphere Reserve.

Additional Information and Declarations

Competing Interests

Author Contributions

Field Study Permissions

Data Availability

The authors declare there are no competing interests.

Othón Alcántara-Ayala conceived and designed the experiments, performed the experiments, analyzed the data, prepared figures and/or tables, authored or reviewed drafts of the paper, approved the final draft.

Ken Oyama conceived and designed the experiments, analyzed the data, contributed reagents/materials/analysis tools, authored or reviewed drafts of the paper, approved the final draft.

César A. Ríos-Muñoz and Santiago Ramirez-Barahona analyzed the data, authored or reviewed drafts of the paper, approved the final draft.

Gerardo Rivas analyzed the data, prepared figures and/or tables, authored or reviewed drafts of the paper, approved the final draft.

Isolda Luna-Vega conceived and designed the experiments, performed the experiments, contributed reagents/materials/analysis tools, prepared figures and/or tables, authored or reviewed drafts of the paper, approved the final draft.

The following information was supplied relating to field study approvals (i.e., approving body and any reference numbers):

A field permit of scientific collection (Official number SGPA / DGVS / 12770/16) was issued by the Secretaria de Medio Ambiente y Recursos Naturales, of Mexico.

The following information was supplied regarding data availability:

The raw data are available in the Supplemental Files.

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
