# Peer review of "Morphological variation of leaf traits in the Ternstroemia lineata species complex (Ericales: Penthaphylacaceae) in response to geographic and climatic variation"

_PeerJ, doi:10.7717/peerj.8307_

## Round 0.1 · original submission · Major Revisions

Dear Authors,

This manuscript provides a large dataset and interesting insights on leaf morphological variation in Ternstroemia lineata species complex across a large spatial scale in Mexico and Guatemala.

Reviewers were positive on the potential for publication. However the manuscript as it stands needs to be revised in some specific points highlighted below:

1. In the introduction, improve structure and provide clear research questions, expectations and background for them. The role of biogeographic provinces, geographic units and subspecies in the study is not clear. Explain the importance of them to understand leaf morphological variation and make it clear which variables are part of your questions.
2. Consider showing simple direct relationships, e.g. scatterplots, between leaf morphological variables and climatic variables (or other way that present a clear visual relationship). RDA graphs are not very helpful for that.
3. If subspecies identification is available, present the sampling effort distribution among them and consider subspecies on the analyses or at least on the discussion of the results.
3. Give more focus on the relationship between leaf morphological variation and climate than with geographical regions, so the results have more generality.
4. Address carefully every other specific point highlighted by the reviewers.

If the authors provide a new version of the manuscript that incorporates these points and reviewers concerns they will have an improved version of it potentially being a well worth paper to be published in PeerJ.

Sincerely,

Juliana Schietti

Reviewer 1 ·

Basic reporting

Line 75, leaf size is also differed within a canopy with the top leaves being smaller and more thick. And in this paragraph, when authors talking about leaf size or "Smaller leaves are better adapted to hot or dry environments", does it refer to broad-leaf or even to conifer needles?

Line 75, it is hard to apart the cause due to precipitation/nutrient content from the cause due to different altitudes, normally precipitation also varied along a gradient. So should this sentence be "...with the increase in altitude because of the differences in precipitation and ?"?

Line 112-118, line 119-125 can be combined as one paragraph. And Line 119-125 should be mentioned before line 112-118 since it discussed the distribution of the species (more broad knowledge).

Line 129, the second objective is not a really objective, it is a discussion point. Unless authors conducted some meta-data analysis, this does not count as an objective.

Experimental design

Line 136-145, did authors measure those variables in the field or in the lab? If it was done in the lab, more details should be mentioned about how to preserve samples, fix samples on a white background, etc...Since authors used vernier to measure the size, I don't see any needs to collect leaves, they can just measure directly in the field.

Line 136, did authors record the locations of each sampled individuals using GPS, so later on they could map them?

Line 183, "ArcGIS"

In the Method section, for sentences using GIS, it might be better to attach a flowchart (for example if using a model builder), so readers would better understand the methods.

Validity of the findings

Line 217, no need to use "whereas", use "and". this is not a contrasting

Line 230, "Spatulated vs. elliptic leaves (DWLR)." this is not a complete sentence. Is this a subheading?

Line 249, I did not see any information of percent error in Table 4, how much is the minimum?

Line 259, typo "for de first"

Line 268, Were there any correlation between geography and climate in spatial? I think they are related. The reason that authors found climate is better indicate leaf shape than geography might be due to the small variance of climate (spatial) at this range. Right now, authors is comparing climate (seasonal/temporal) to geography (spatial).

Line 295, environmental factors due to gradients?

Line 298-302, those three sentences already appeared in the Result sections (line 230). Authors should not repeat the findings again but add more explains/linkage about environmental factors and leaves (spatulate vs. elliptic). Sentences in line 317 should be brought up here.

Line 350-354, was it necessary to conclude those methods in the Conclusion section? If those are new techniques that researchers have rarely applied to this area, it might be worth to mention. But I don't think interpolation map is.

Additional comments

In this study, authors measured 9 foliar traits from 270 individuals of Ternstroemia lineata complex, and related 19 environmental variables to variation in leaf morphology across a range in Mexico and north of Guatemala. I find this paper add some values to the society, however authors should reorganize the Discussion as the Discussion does not have to follow the order in result section. For example, it is better to group sentences in line 292-302 and in line 317-321 because they both talking about "spatulate leaf shape". Other comments please see above.

Reviewer 2 ·

Basic reporting

This is a very interesting study that evaluated the intraspecific variation on leaf traits of Ternstroemia lineata in relation to climate and geographical variation. For this, the authors compared nine morphological leaf traits of 270 individuals of 32 populations from different regions of Mexico, delimiting groups and boundary zones of leaf morphology changes. Also, they evaluated which environmental variables influence more traits variability. I think this is a nice work with potential to be published, although it needs some improvements. The English grammar needs some revision. Some parts are not easy to follow and I found some words in Spanish. The whole text needs to be reviewed, especially on the results and discussion. The tables can be formatted to be more presentable and some figures needs to be redone.

Experimental design

I missed some information on the methods. There is no description regarding the environmental differences between the areas (provinces), so I do not know how different they are and I cannot associate them with the results. There is no information of how many samples per province were taken. I can find it in the supplementary material, but I think it should be described on the text. The Figure 1 is also not illustrative of how many samples per area they have. Looking at the supplementary material, I can see that the number of populations per area is not equal. It can influence on the results and should be clear on the text. I’m not sure if the use of three RDA analysis is necessary and brings additional information.

Validity of the findings

The results are interesting, but need to be better described and discussed. For example, there are nine areas that differed in leaf morphology, but there is no description of them. There is a description of the variation of the three traits selected (TL, WLR, DWLR), but I do not understood how these nine areas differ in leaf morphology. In the discussion, there is no information regarding the relation of each leaf characteristic with the environmental variables, for example, why leaves grow more on higher temperatures. The authors describe that larger leaves are related to higher temperatures, but do not say why it could be an important adaptation. Which are the adaptive advantages to have larger or shorter leaves, more spatulate leaves or elliptic leaves, in relation to the environmental variables analyzed? They briefly presented these relationship in the introduction, but it is not well explored. If the aim of the paper is to evaluate how traits plasticity in response to environmental characteristics can lead to adaptive evolution of species, it should be discussed. Also, there is no link between leaf traits with the subspecies. You discussed some things on the conclusions that you didn’t discussed before. Here you should summarize what are your main results, contributions of the study and which is still necessary to explore.

Additional comments

Line 34: ratio of the leave? You mean the ratio between the maximum width of the lamina and the lamina length (WLR)
Line 40-44: rewrite.
Line 61: What you mean with ‘current ecology’?
Line 68: you mean local climate?
Line 71: appreciate? Suggestion: to observe
Line 73: to the plant
Lines 78-79: This phrase is strange. And how does light adjust leaf shape?
Line 91: “…. key climatic factors to which it is related have been identified” I would remove ‘have been identified’.
Line 103: How they are correlated? Which biotic factors?
Lines 106-107: I think the transition of subject was very abrupt
Line 128: aim is better than objective
Line 151: “With these selected morphological... “ , here you should indicate which traits were chosen with the PCA, not in line 154.
Lines 155-158: I think that the explanation of what the ratios WLR e DWLR indicate should be together with the descriptions of the traits, on the earlier topic. This topic in more data analysis.
Line 224: visual inspection?
Lines 230-237: Rewrite this sentence, and follow a pattern to describe the results. It will be interesting if you remember the writer what means a higher or lower DWLR. Same thing for WLR.
Lines 245: It is better ‘discriminant analysis’. You don’t need to repeat that the nine groups are based on the three traits selected, because it is already explained.
Lines 252-254: rewrite this sentence. Suggestion: Based on iterative analysis we selected the environmental variables: ……
Lines 307-308: this phrase is confuse, I didn’t understand.
Lines 308-309: Most important variables for what?
Lines 314-315: clear correlation means positive or negative?
Lines 322-325: I didn’t understand.
Line 339: you didn’t mentioned before different levels of response
Line 342: appreciated?
You discussed some things on the conclusions that you didn’t discussed before. Here you should summarize what are your main results, contributions of the study and which is still necessary to explore.

Figure 1. It is not possible to see every point on the map, they overlap a lot. And the same thing happen to the ‘known distribution area’.
Figure 5. The canonical distribution graphic is too small.

Reviewer 3 ·

Basic reporting

The work is important and trying to understand relationship among species and climate a important subject in face of the warm up of the planet.

The article is quite well written, though it is not always clear and unambiguous. It would benefit from a thorough review of the English.

Abstract – it provides a complete abstract showing the importance of the study, however the results are spatially described by Mexico areas (Soconusco, Sierra Madre del Sur, Eje Volcanico province), rather than their spatial ecological gradient, making it difficult for the reader to understand the main result (if they are not from Mexico). Please check Hans ter Steege et al. in 2006 (Nature paper) that found major gradients in Amazonia, for ideas on how to present your results in the abstract.


The Introduction introduces some of the main contextual issues but is not very well structured, with the narrative jumping between conceptual areas without very strong links. Also, the complexity of interactions that mould the relationship between leaf morphology and environmental conditions is not well explained, with no mention of the numerous trade-offs that can influence multiple aspects of a plant´s phenotype.

Also, there are a lot of debate about using “lat/long” as variable in this type of analyses, the introduction list some work that focus on this, but does not take a path in the argumentation.

Maybe it would be good to explain what this species T Lineata is an important species to be studied (see the work of Stropp at all 2017 in Oecologia)

It would also have been nice to have seen a synthesis of the methodological challenges of this type of study, such as how to deal with spatial autocorrelation and whether or not to control for phylogeny.

Finally, the article does not provide clear predictions for how leaf morphology in the study may change in relation to the environmental gradient. Why we think this study maybe bring new insights into the subject? Or new insights for a new ecosystem, etc.

The Figures are relevant, high quality, well labelled & described. The raw data is supplied

Experimental design

This represents original primary research that is clearly within Scope of the journal. However, the research question could be better defined (see above) and other than providing another example of phenotypic change over an environmental gradient it is not clear how the research is filling an important knowledge gap. On the positive side, the investigation appears to be performed to high technical and ethical standards and the methods were described with sufficient detail and information to replicate.

From the statistical perspective, it is not clear how the authors dealt with spatial autocorrelation and whether there was a need to control for the phylogenetic relationships between sub-populations within the species complex. Perhaps this information was not available?

Validity of the findings

Results start with “ the PCA explains” …… I really wanted to find out more about your measurements, sizes, means, description of your data before we know about the results of the PCA.

The results are a bit difficult to interpret, maybe for this type of study would be good to have a clear hypothesis? I understand that the new framing of research for big data, we do not need hypothesis to test, but in this case, I think it might make the paper more focused.

Given the large number of environmental variables and high sample size, maybe we could expect that some strong relationships with leaf traits would appear. However, it would be nice to see some (convincing) adaptive explanations for why certain traits (e.g. leaf length) appear to have strong correlations with particular bioclimatic variables – while others do not.

The geographic variation in leaf morphology is fascinating and real, but it is still not clear that the research have i) identified the main drivers of this variation, and ii) excluded/controlled for the effects of phenotypic plasticity and phylogeny/biogeography.

Additional comments

This is a well-researched and detailed study that generates a large and interesting database. However, the research question could be better framed and the analysis might not be adequate to identify the key drivers of the observed geographic patterns of phenotypic variation in leaves.

---

## Round 0.2 · Minor Revisions

My apologies for the delay in returning on the revised manuscript. The reviewers made their contributions on time but it came to me at a very busy time when I teach intensive courses.

The manuscript has improved and is likely to be published with some further adjustments suggested by the reviewers.

Reviewer 1 ·

Basic reporting

None

Experimental design

Line 200, what is the linear correlation? Pearson correlation or Spearman?

Validity of the findings

None

Additional comments

I appreciated author's hardworking on revising the manuscript. New analysis was added in, and both Introduction and Discussion were improved a lot.

Reviewer 2 ·

Basic reporting

The authors did a very nice work including all the reviewers’ suggestions and I indicate it for publication at PeerJ. I have just few considerations that are detailed below and correted some mistakes in the main text.

Line 74-75: For instance, leaf size has been shown to decrease with increasing altitude because of the differences in precipitation and nutrient content in the soil decrease
Suggestion: For instance, leaf size has been shown to decrease with increasing altitude because of the differences in precipitation and with the decrease of soil nutrient content.
Line 112: “cloudy conditions” is better
Line 118: suggestion: We analyzed how leaf morphology of the Ternstroemia lineata species complex vary in relation to climate variables across its geographic distribution to detect which environmental variables have influenced the leaf morphological differentiation in this species complex.
Line: 130: “… herbarium specimens identification?
Line 362 and 363: cloud or mist is better than fog
Line 364: transpiration is better than perspiration
Conclusions: You can make it more simple, going direct to the point and avoiding phrases like “in this study, we analyzed…..” or “our analysis showed…”
Graphics: Write the variable names in the correlation graphics, not the abbreviation. The number of the axis are too small.

Experimental design

no comment

Validity of the findings

no comment

Additional comments

no comment

Annotated reviews are not available for download in order to protect the identity of reviewers who chose to remain anonymous.

---

## Round 0.3 · accepted · Accept

The authors have responded to the reviewers' final comments, and I am pleased to inform that your manuscript is scientifically suitable for publication. Congratulations!